# A Mixture of Algae and Extra Virgin Olive Oils Attenuates the Cardiometabolic Alterations Associated with Aging in Male Wistar Rats

**DOI:** 10.3390/antiox9060483

**Published:** 2020-06-03

**Authors:** Daniel González-Hedström, Sara Amor, María de la Fuente-Fernández, Antonio Tejera-Muñoz, Teresa Priego, Ana Isabel Martín, Asunción López-Calderón, Antonio Manuel Inarejos-García, Ángel Luís García-Villalón, Miriam Granado

**Affiliations:** 1Departamento de Fisiología, Facultad de Medicina, Universidad Autónoma de Madrid, 28029 Madrid, Spain; dgonzalez@pharmactive.eu (D.G.-H.); sara.amor@uam.es (S.A.); maria.delafuente@uam.es (M.d.l.F.-F.); antonio.tejera@quironsalud.es (A.T.-M.); angeluis.villalon@uam.es (Á.L.G.-V.); 2Pharmactive Biotech Products S.L. Parque Científico de Madrid, Avenida del Doctor Severo Ochoa, 37 Local 4J, Alcobendas, 28108 Madrid, Spain; aminarejos@hotmail.com; 3Departamento de Fisiología, Facultad de Medicina, Universidad Complutense de Madrid, 28040 Madrid, Spain; tpriegoc@med.ucm.es (T.P.); anabelmartin@med.ucm.es (A.I.M.); alc@med.ucm.es (A.L.-C.); 4CIBER Fisiopatología de la Obesidad y Nutrición, Instituto de Salud Carlos III, 28029 Madrid, Spain

**Keywords:** aging, omega 3 fatty acids, extra virgin olive oil, insulin resistance, cardiovascular, inflammation, oxidative stress, endothelial dysfunction

## Abstract

Aging is one of the major risk factors for suffering cardiovascular and metabolic diseases. Due to the increase in life expectancy, there is a strong interest in the search for anti-aging strategies to treat and prevent these aging-induced disorders. Both omega 3 polyunsaturated fatty acids (ω-3 PUFA) and extra virgin olive oil (EVOO) exert numerous metabolic and cardiovascular benefits in the elderly. In addition, EVOO constitutes an interesting ingredient to stabilize ω-3 PUFA and decrease their oxidation process due to its high content in antioxidant compounds. ω-3 PUFA are commonly obtained from fish. However, more ecological and sustainable sources, such as algae oil (AO) can also be used. In this study, we aimed to study the possible beneficial effect of an oil mixture composed by EVOO (75%) and AO (25%) rich in ω-3 PUFA (35% docosahexaenoic acid (DHA) and 20% eicosapentaenoic acid (EPA)) on the cardiometabolic alterations associated with aging. For this purpose; young (three months old) and old (24 months old) male Wistar rats were treated with vehicle or with the oil mixture (2.5 mL/kg) for 21 days. Treatment with the oil mixture prevented the aging-induced increase in the serum levels of saturated fatty acids (SFA) and the aging-induced decrease in the serum concentrations of mono-unsaturated fatty acids (MUFA). Old treated rats showed increased serum concentrations of EPA and DHA and decreased HOMA-IR index and circulating levels of total cholesterol, insulin and IL-6. Treatment with the oil mixture increased the mRNA levels of antioxidant and insulin sensitivity-related enzymes, as well as reduced the gene expression of pro-inflammatory markers in the liver and in cardiac and aortic tissues. In addition, the treatment also prevented the aging-induced endothelial dysfunction and vascular insulin resistance through activation of the PI3K/Akt pathway. Moreover, aortic rings from old rats treated with the oil mixture showed a decreased response to the vasoconstrictor AngII. In conclusion, treatment with a mixture of EVOO and AO improves the lipid profile, insulin sensitivity and vascular function in aged rats and decreases aging-induced inflammation and oxidative stress in the liver, and in the cardiovascular system. Thus, it could be an interesting strategy to deal with cardiometabolic alterations associated with aging.

## 1. Introduction

The progressive increase in the number of elderly persons in the society represents a major sanitary and economic problem, due to the rise in age-related disorders, particularly metabolic and cardiovascular diseases [1]. For this reason, there is a strong interest in developing new strategies and procedures to slow down the incidence of these alterations.

There is evidence that caloric restriction improves both metabolic and cardiovascular health in aging [2,3]. However, due to the difficulties in implementing this procedure in the population and the lack of effective pharmacological treatments to decrease body weight, other dietary interventions, that may have an equivalent benefit, are under scrutiny. Among them, a diet enriched in fish-derived omega-3 fatty acids (ω-3 PUFA) has been found to promote cardiovascular and neurological health throughout life [4]. Particularly, supplementation with ω-3 PUFA improves endothelial function in healthy subjects [5] as well as in aged [6], hypercholesterolemic [7] or type 2 diabetic patients [8].

Fish oil also prevents insulin resistance [9], which is one of the most common features of the aging-induced metabolic disorders [10]. Moreover, decreased insulin sensitivity, not only affects metabolic tissues, but also cardiovascular function [11] by reducing insulin-induced arterial vasodilation [12] and cardiac contractility [13]. Our group has previously reported that the cardiovascular insulin resistance associated with aging is partially prevented by a moderate protocol of caloric restriction [12,13]. In blood vessels, caloric restriction improves aging-induced vascular insulin resistance by reducing inflammation and enhancing nitric oxide release trough the activation of the phosphatidylinositol 3 kinase (PI3K)/Akt pathway [12]. However, there is no evidence on whether supplementation with a mixture of EVOO and AO rich in ω-3 PUFA are also beneficial in this particular alteration.

The consumption of ω-3 PUFA from wild fish or fish oil is limited and not very sustainable in the long-term [14]. Aquaculture, or fish farming, offers an alternative, although this practice has been associated with damage and contamination of ecological areas [15]. For this reason, oil from marine algae has been proposed as an alternative and more sustainable source of ω-3 PUFA [16,17].

Although there are already some studies showing the benefits of the consumption of oils extracted from algae on inflammatory processes [18] and cardiovascular health [19], there is no research which systematically covers the effects of these oils on the metabolic and cardiovascular alterations associated with aging.

An important issue regarding ω-3 PUFA is that they have a high susceptibility to oxidation, which makes necessary technological strategies such as the addition of antioxidant compounds to improve their stability [20]. In this regard, extra virgin olive constitutes an ideal ingredient to stabilize the ω-3 PUFA from marine algae oils due to the vast diversity of antioxidant substances that it contains, mainly phenolic compounds [21]. Furthermore, addition of EVOO to AO is attractive, not only from a technological point of view due to its usefulness in the stabilization of ω-3 PUFA, but also because it exerts several beneficial effects on the human health, especially in a context in which inflammation and cellular oxidative damage are increased [22]. Indeed, olive oil intake is associated with decreased risk of overall mortality [23], and has been negatively associated with cardiovascular risk [24] as it reduces the development of atherosclerosis through the reduction of pro-inflammatory markers [25]. Some of these benefits are due to monounsaturated fatty acids present in the saponifiable fraction, mainly to oleic acid (55–83%), that is reported to diminish inflammation and improve diastolic function in mice [26], as well as reduce the atherosclerotic process [27,28] and cardiovascular insulin resistance in vitro [28].

However, the beneficial cardiometabolic effects of olive oil are also due to other minority compounds (about 2% of total) within the non-saponifiable fraction, such as squalene, polyphenols (mainly tyrosol, hydroxytyrosol, and derivatives) and vitamins (α, β, γ and δ tocopherols) [29,30]. These compounds improve the lipid profile and reduce the attachment of cell adhesion molecules in early atherogenesis and are reported to exert antioxidant, anti-inflammatory, anti-fibrotic and anti-obesity effects among others [29].

Based on this background we hypothesized that a mixture of both AO rich in ω-3 PUFA and EVOO rich in oleic acid and polyphenols may be useful for the treatment/prevention of the cardiometabolic alterations associated with aging. Therefore, the aim of this study was to analyze the possible beneficial effects of a mixture of EVOO and AO in a ratio 3:1 on the age-induced metabolic and vascular alterations, especially in the aging-induced vascular insulin resistance.

## 2. Material and Methods

### 2.1. Animals

Young rats (3 months old; *n* = 11) and old (24 months old; *n* = 13) male Wistar rats were housed under controlled conditions of humidity (50–60%) and temperature (22–24 °C). A standard chow was provided *ad libitum* to all animals. All the experiments were performed following the European Union Legislation and with the approval of the Animal Care Committee of the Universidad Autónoma de Madrid and the Autonomic Goverment (PROEX 048/18).

### 2.2. Treatment

Half of the old rats (Old + Oil Mixture) were treated once daily by gavage (intragastric tube) with 2.5 mL/kg of a mixture of 75% of EVOO (Cornicabra variety; 80% oleic acid and 63.49 mg/g of secoiridoids) (Aceites Toledo S.A., Los Yébenes, Spain) and 25% of AO (Schizochytrium spp: 35% DHA, 20% EPA and 5% Docosapentaenoic (DPA)) (DSM, Heerlen, Netherlands) for 21 days. Young animals and the other half of old rats and received tap water by gavage once daily (2.5 mL/kg). This proportion of AO and EVOO was used according to previous studies, which demonstrate that it significantly reduces the oxidative degradation of ω-3 PUFA [31]. 

A daily control of body weight and a weekly control of food and water intake was performed over the three-week treatment. At the end of the treatment, the animals were killed after overnight fasting by injection of an overdose of sodium pentobarbital (100 mg/kg) and decapitation. After decapitation blood was collected and centrifuged (20 min at 3000 rpm) to collect the serum. Visceral (epididymal), subcutaneous (lumbar), brown (interscapular) and perivascular (aortic) adipose tissue depots, as well as kidneys, adrenal glands, spleen, liver and heart were immediately removed and weighed. All tissues were stored at −80 °C for later analysis.

### 2.3. Serum Measurements

(a) Metabolic Hormones

ELISA kits (Merck Millipore, Dramstadt, Germany) were used to measure serum concentrations of leptin, insulin and adiponectin according to the instructions of the manufacturer. Sensitivity and intrassay variation were 0.04 ng/mL and 1.9–2.5% for leptin assay, 0.2 ng/mL and 1.9–2.5% for insulin and 0.16 ng/mL and 0.43–1.96% for adiponectin.

(b) Lipid profile

Commercial kits from Spin React S.A.U (Sant Esteve de Bas, Gerona, Spain) were used to measure serum triglycerides, total lipids, total cholesterol, low-density lipoprotein (LDL), and high-density lipoprotein (HDL) according to the instructions of the manufacturer.

(c) Pro-inflammatory mediators

ELISA kits (Cusabio, Wuhan, China) were used to measure plasma levels of interleukin-6 (IL-6) and tumor necrosis factor alpha (TNF-α) according to the instructions of the manufacturer. These kits have a sensitivity of 0.078 pg/mL (IL-6) and 1.56 pg/mL (TNF-α), and an intraassay variation of <8% and interassay variation of <10% for both assays.

### 2.4. Serum Lipid Extraction and Fatty Acid Analysis

The Fatty Acid (FA) profile was determined by gas chromatography (GC). Serum samples were processed following the lipid extraction procedure described by Drews et al. 2018 [32]. Briefly, to 25 µL of each serum sample 500 µL of methanol:MTBE (Methyl tert-butyl ether):cholorform 1.33:1:1 *v*/*v*/*v* were added. The mixture was incubated at 23 °C for 30 min and then centrifuged at 13,200 rpm for 10 min. Subsequently, the supernatants were evaporated to dryness at 40 °C. The obtained residues were methylated by 0.2 mL of toluene, 1.5 mL of methanol (MeOH) and 0.3 mL of 8% (*w*/*v*) HCl (in 85:15 MeOH:H_2_O *v*/*v* which gave a final HCl concentration of 0.39 M). The mixture was incubated overnight at 45 °C. After that, 1 mL of hexane and 1 mL of H_2_O were added, and the samples were vortexed 3 s 10 times. The upper phases of hexane were obtained with a 60 s centrifugation at 2000 rpm. Samples were then subjected to the GC protocol.

The identification and quantification of FA was performed on a Shimadzu GC-2025 gas chromatograph (Shimadzu; Kioto, Japan) with an AOC-20i autosampler (Shimadzu; Kioto, Japan) and a FID detector, equipped with a ZB-FFAP capillary column of 50 m length, 0.32 mm i.d., and 0.50 µm film thickness (Phenomenex; Torrance, CA, USA). The carrier gas was Helium at a flow rate of 1.1 mL/min, with temperatures of 250 °C for both injector and detector. The injecting sample volume was 1 µL. The GC method was performed using the following temperature program: start at injection 50 °C, hold for 1 min, followed by a 15 °C/min oven temperature ramp to 230 °C; hold for 3 min, followed by a 10 °C/min ramp to 250 °C [33]. To identify the FA peaks the known fatty acid mix Supelco 37 Component FAME Mix (Merck; Dramstadt, Germany) was used as a standard. The levels of SFA were obtained by the sum of palmitic, palmitoleic and stearic acids. Likewise, the levels of MUFA were obtained by the sum of oleic and linoleic acids, and the levels of PUFA by the sum of ALA, EPA and DHA.

### 2.5. Experiments of Vascular Reactivity

After sacrifice, the aorta was dissected and cut into segments of 2 mm-long in sterile cold saline solution (NaCl 9 g/L). Afterwards, two steel wires with a diameter of 100 µm were passed through the lumen of each arterial segment. Then, one of the wires was fixed to a 4-mL organ bath containing modified Krebs-Henseleit solution with the following composition (mM): NaCl, 115; KCl, 4.6; KH_2_PO4, 1.2; MgSO_4_, 1.2; CaCl_2_, 2.5; NaHCO_3_, 25; glucose, 11. The solution within the organ bath was kept at 37 °C and the pH was adjusted to 7.3–7.4 by equilibration with 95% oxygen and 5% carbon dioxide. The other wire was connected to a strain gauge (Universal Transducing Cell UC3 and Statham Microscale Accessory UL5, Statham Instruments, Inc.; Rochester, NY, USA), in order to measure the isometric tension with a PowerLab data acquisition system (AD Instruments, Colorado Springs, CO, USA). The segments were equilibrated at an optimal passive tension of 1g during 60–90 min and afterwards they were stimulated with potassium chloride (KCl; 100 mM) to assess the ability of vascular of smooth muscle to contract. Only segments that contracted at least 0.5 g to KCl were used.

The vasoconstrictor response to accumulative doses of noradrenaline (10^−9^–10^−4^ M), endothelin-1 (ET-1) (10^−9^–10^−7^ M) or angiotensin-II (10^−11^–10^−6^ M) was recorded in abdominal aortic segments. Results were expressed as percentage of the contraction to KCl 100 mM.

To study the vasodilator response, thoracic aortic segments were pre-contracted with phenylephrine 10^−7.5^ M to subsequently perform a cumulative dose-response curve in response to acetylcholine (10^−9^–10^−4^ M), sodium-nitroprusside (10^−9^–10^−5^ M) or insulin (10^−8^–10^−5^ M). Relaxation was expressed as percentage of the initial tone.

All drugs were obtained from Sigma-Aldrich, (St. Louis, MO, USA).

### 2.6. Incubation of Aorta Segments in Presence/Absence of Insulin (10^−7^ M):

The 2 mm-long segments from the thoracic aorta were placed in 6 well culture plates, containing 1.5 mL of Dulbecco’s Modified Eagle’s Medium and Ham’s F-12 medium (DMEM/F-12) with glutamine from Gibco (1:1 mix; Invitrogen, Carlsbad, CA, USA), supplemented with 100 U/mL penicillin and 100 μg/mL streptomycin (Invitrogen, Carlsbad, CA, USA). The 3 segments/well were incubated at 37 °C in a 95% O_2_ and 5% CO_2_ incubator for 30 min with insulin (10^−7^ M) (Sigma-Aldrich, St. Louis, MO, USA) or vehicle. After incubation, the segments and the culture media were collected and stored at −80 °C for further studies.

### 2.7. Nitrite and Nitrate Concentrations in the Culture Medium:

Nitrite and nitrate concentrations were measured in the culture medium from aorta segment incubations by a modified method of the Griess assay [34]. Briefly, 100 μL of vanadium chloride (Sigma-Aldrich, St. Louis, MO, USA) were added to 100 μL of culture medium on a 96-well plate. Immediately after, 100 μL of the Griess reagent (1:1 mixture of 1% sulfanilamide (Merck Millipore, Darmstadt, Germany), and 0.1% naphthylethylenediamine dihydrochloride (Merck Millipore, Darmstadt, Germany)) were added to each well and incubated at 37 °C for 30 min. After incubation, absorbance was measured at 540 nm. Nitrite and nitrate concentrations were calculated using a NaNO_2_ standard curve and was expressed in µM.

### 2.8. Protein Analysis in Arterial Tissue by Western Blot

The 100 mg of arterial tissue were homogenized using RIPA buffer and the total protein content in each sample was assessed by the Bradford method (Bradford 1976). Proteins were separated by electrophoresis in sodium dodecyl sulfate (SDS) acrylamide gels (10%) after loading 100 μg of protein in each well. After the electrophoresis, proteins were transferred to polyvinylidine difluoride (PVDF) membranes (Bio-Rad, Hércules, CA, USA). The transfer efficiency was assessed by Ponceau red dyeing (Sigma-Aldrich, St. Louis, MO, USA). Membranes were then blocked with Tris-buffered saline (TBS) containing 5% (*w*/*v*) non-fat dried milk for 2h. Afterwards, membranes were incubated with the primary antibody for Akt (1:1000; #04-796, Merck Millipore; Dramstadt, Alemania) and p-Akt (Ser473) (1:500; #9271, Cell Signaling Technology; Danvers, MA, EE.UU.) at 4 °C overnight. After incubation, the membranes were washed three times with TTBS (tris-buffered saline (TBS) + Tween 0.1%) and incubated with the secondary antibody conjugated with peroxidase (1:2000; Pierce, Rockford, IL, USA). After a 90 min-incubation, the peroxidase activity was visualized by chemiluminescence using BioRad Molecular Imager ChemiDoc XRS System. All data were normalized to the housekeeping protein GAPDH (Thermo Fisher Scientific, Hampton, NH, USA) and referred to % of control values (samples from young rats) on each gel.

### 2.9. RNA Extraction and Purification

Total RNA was extracted from myocardial, liver and arterial tissue according to the Tri-Reagent protocol [35] and quantified with Nanodrop 2000 (Thermo Fisher Scientific, Hampton, NH, USA). cDNA was then synthesized from 1 µg of total RNA using a high capacity cDNA reverse transcription kit (Applied Biosystems, Foster City, CA, USA).

### 2.10. Quantitative Real-Time PCR

Assay-on-demand kits (Applied Biosystems, Foster City, CA, USA) were used for quantitative real-time polymerase chain reaction (qPCR) to determine the gene expression of interleukin 1β (IL-1β, Rn00580432_m1), interleukin 6 (IL-6, Rn01489669_m1), interleukin 10 (IL-10, Rn01483988_g1), tumoral necrosis factor alpha (TNF-α, Rn01525859_g1), ciclooxigenase-2 (COX-2, Rn01483828_m1), inducible Nitric Oxide Synthase (iNOS, Rn00561646_m1), NADPH oxidase 1 (NOX1, n00586652_m1), NADPH oxidase 4 (NOX4, Rn00585380_m1), glutathione reductase (GSR, Rn01482159_m1), glutathione peroxidase (GPx, Rn00574703_m1), glucokinase (GCK, Rn00561265_m1), glycogen synthase kinase 3β (GSK3β, Rn01444108_m1), superoxide dismutase-1 (SOD1, Rn00566938_m1) and lipoxygenase (Alox5, Rn00563172_m1) in myocardial, liver and arterial tissues. Amplification was performed with TaqMan Universal PCR Master Mix (Applied Biosystems, Foster City, CA, USA) in a Step One machine (Applied Biosystems, Foster City, CA, USA). Values were normalized to the housekeeping gene 18S (Rn01428915_g1). For each gene, the ∆∆CT method was used to determine the relative expression levels [36]. The values were referred to % of controls (samples from young rats).

### 2.11. Statistical Analysis

The values are expressed as means ± standard error of the mean (SEM) and analyzed by one-way ANOVA followed by Bonferroni post-hoc test using GraphPad Prism 5.0. (San Diego, CA, USA). A *p* value of < 0.05 was considered significant.

## 3. Results

### 3.1. Body Weight and Food Intake

Body weight gain and daily food intake over the 21-day treatment are shown in Figure 1. Young rats gained weight whereas old rats lost weight (*p* < 0.001). Old rats treated with oil mixture also lost weight, but to a less extent than untreated rats (*p* < 0.01). Caloric intake was unchanged between young and untreated old rats and significantly reduced in old rats treated with the oil mixture compared to young animals (*p* < 0.05).

### 3.2. Organ Weights

Table 1 shows the organ weights from 3 and 24 months-old rats, treated with vehicle or with the oil mixture for 21 days. Aging was associated with an increased weight of heart (*p* < 0.01), epidydimal visceral adipose tissue (*p* < 0.001), lumbar subcutaneous adipose tissue (*p* < 0.01), interscapular brown adipose tissue (*p* < 0.05), periaortic adipose tissue (*p* < 0.05), kidneys (*p* < 0.01), adrenal glands (*p* < 0.05), liver (*p* < 0.05) and spleen (*p* < 0.001). The aging-induce increase in the weights of heart, kidneys, adrenal glands, liver and spleen was prevented old rats treated with the oil mixture (*p* < 0.05 for all).

### 3.3. Lipid Profile, Serum Levels of Metabolic Hormones and HOMA-IR Index

Old rats showed increased serum levels of total lipids (*p* < 0.05), triglycerides (*p* < 0.05), total cholesterol (*p* < 0.05), LDL-cholesterol (*p* < 0.01), insulin (*p* < 0.01), leptin (*p* < 0.01) and adiponectin (*p* < 0.001), as well as higher HOMA-Index (*p* < 0.05) compared to young rats. Treatment with the oil mixture prevented the aging-induced increase in total lipids (*p* < 0.05), triglycerides (*p* < 0.05), total cholesterol (*p* < 0.001), LDL-cholesterol (*p* < 0.001), insulin (*p* < 0.05) and HOMA-Index (*p* < 0.05) (Table 2).

### 3.4. Serum Inflammatory Parameters

In old rats the circulating levels of IL-6 and TNFα were significantly increased compared to young rats (*p* < 0.05 for both), and both pro-inflammatory cytokines were reduced in the serum of rats treated with the oil mix (*p* < 0.05) (Table 2).

### 3.5. Percentage of Fatty Acids in the Serum

Aging was associated with a significant increase in the % of SFA and with a reduction in the % of the MUFA (*p* < 0.01). Treatment with the oil mixture significantly decreased the % of the SFA and increased the % of MUFA compared to untreated old rats (*p* < 0.05 for all). The % of PUFA increased compared to both young and old rats administered with vehicle (*p* < 0.05 for both) (Table 3).

There were not changes in the percentage of stearic and linoleic acids in the serum, but the percentage of saturated palmitic and palmitoleic acids was increased (*p* < 0.05), and that of monounsaturated oleic acid and polyunsaturated ALA was reduced (*p* < 0.01) by aging.

The percentage of palmitoleic acid was higher in old rats treated with the oil mixture compared to young animals (*p* < 0.01). However, the percentage of palmitic acid was not different between young and old rats treated with the oil mixture. Treatment with the oil mixture increased the % of oleic acid in old rats (*p* < 0.05) although these levels remained lower than those in young rats (*p* < 0.05). Finally, treatment with the oil mixture significantly increased the % of DHA compared to untreated old rats (*p* < 0.05) and the % of EPA, compared with both young and old rats, administered with vehicle (*p* < 0.05 for both) (Table 3).

### 3.6. Gene Expression of Metabolic, Inflammatory and Oxidative Stress Markers in the Liver

The hepatic gene expression of the enzymes GCK and GSK3β was reduced in old rats (*p* < 0.05) and increased by treatment with the oil mixture (*p* < 0.05) (Figure 2).

In the liver, the gene expression of the pro-inflammatory markers iNOS (*p* < 0.01) and TNFα (*p* < 0.05) were significantly increased in old rats and reverted by oil mixture treatment (*p* < 0.05). In addition, the treatment with the oil mixture also reduced mRNA levels of COX-2 (*p* < 0.01) and IL-1β (*p* < 0.05) compared with untreated old rats (Figure 2).

Aging was associated with a significant reduction in the hepatic gene expression of the antioxidant enzymes GSR and SOD-1 (*p* < 0.05, and *p* < 0.001, respectively). Likewise, the expression of NOX-4 (*p* < 0.001) and Alox5 (*p* < 0.01) were also significantly reduced. Treatment with the oil mixture did not affect the gene expression of NOX-4 and Alox5 but it prevented the aging-induced reduction of GSR (*p* < 0.01) and SOD-1 (*p* < 0.05) (Figure 1).

### 3.7. Gene Expression of Inflammatory and Oxidative Stress Markers in the Heart

The gene expression of iNOS was significantly increased in the hearts of old rats (*p* < 0.05) and this increase was reverted by treatment with the oil mixture (*p* < 0.01). In addition, the treatment also reduced the mRNA levels of COX-2 (*p* < 0.05), IL-6 (*p* < 0.05) and IL-1β (*p* < 0.05) (Figure 3).

The gene expression of the antioxidant enzymes GSR (*p* < 0.01) and SOD-1 (*p* < 0.001) was downregulated in old compared to young rats whereas the mRNA levels of the pro-oxidant NOX-1 were significantly upregulated (*p* < 0.05). Treatment with the oil mixture prevented the aging-induced reduction in SOD-1 (*p* < 0.05) and the aging-induced increase in NOX-1 (*p* < 0.05). The expression of NOX-4 was reduced (*p* < 0.01) in old rats regardless of whether they had been treated or not with the oil mixture (Figure 3).

### 3.8. Aortic Vasoconstriction

Arterial vasoconstriction, in response to KCl (A), Noradrenaline (B), Endothelin-1 (C) and Angiotensin II (D), are shown in Figure 4. Treatment with oil mixture prevented the aging-induced decrease (*p* < 0.05) in arterial vasoconstriction in response to potassium chloride (*p* < 0.01), but it did not attenuate the aging induced-increased in arterial vasoconstriction in response to noradrenaline.

The contraction to endothelin-1 or angiotensin-II was not modified in old compared to young rats. However, the contraction to angiotensin-II was significantly reduced by treatment with oil mixture (*p* < 0.05) (Figure 4).

### 3.9. Endothelium-Dependent and Independent Aortic Relaxation

The endothelium-independent relaxation in response to sodium nitroprusside (NTP) was not modified in old rats regardless of the treatment (Figure 5). On the contrary, the endothelium-dependent relaxation in response to acetylcholine was significantly reduced in aorta segments from old rats (*p* < 0.01) and partially reverted by oil mixture treatment (*p* < 0.01).

### 3.10. Aortic Relaxation in Response to Insulin

Insulin induced dose-dependent relaxation of aortic segments in all experimental groups. However, this relaxation was significantly reduced in old rats (*p* < 0.05) and improved in old rats treated with the oil mixture (*p* < 0.01) (Figure 6A).

Incubation of aorta segments with insulin increased the protein levels of p-Akt, as well as the release of nitrites to the culture medium in aortas from young (*p* < 0.05) and old rats treated with oil mixture (*p* < 0.05), but not in old rats administered vehicle (Figure 6B,C).

### 3.11. Gene Expression of Inflammatory and Oxidative Stress Markers in the Aorta

The mRNA levels of inflammatory and oxidative stress markers in arterial tissue are shown in Figure 7A, and Figure 7B, respectively.

Aging was associated with an increase in the mRNA levels of COX-2 (*p* < 0.01) and NOX-4 (*p* < 0.001) and with a significant reduction in the gene expression of IL-6 (*p* < 0.01), IL-10 (*p* < 0.05), GPx (*p* < 0.05) and GSR (*p* < 0.05). Treatment with the oil mixture did not modify the aging-induced alterations in the gene expressions of IL-6, GPx and GSR but it significantly attenuated the aging-induced increase in COX-2 (*p* < 0.01) and NOX-4 (*p* < 0.05) mRNA levels, and the decrease in the gene expression of IL-10 (*p* < 0.05).

## 4. Discussion

The results of this study suggest that a mixture of extra virgin olive oil and oil from marine algae rich in ω-3 PUFA may be useful to improve the metabolic and vascular condition in old age. To the best of our knowledge, this is the first study that demonstrates the beneficial effects of a mixture of these oils for the treatment of the cardio-metabolic alterations associated with aging.

In order to assess the possible role of the different fatty acids present in the EVOO and the AO we determined their % in the serum of young and old rats administered vehicle or the oil mixture. Our results show that, as previously described in senescent cells [37], aging was associated with an increase in the serum % of SFA, and with a decrease in the % of MUFA, being these alterations prevented after the treatment with the oil mixture. Likewise, other authors have reported increased SFA plasma levels in aged individuals [38], with this fact being negatively related to insulin sensitivity and positively related to metabolic syndrome and arterial stiffness [39]. In addition, treatment with the oil mixture significantly increased the % of the ω-3 PUFA EPA and DHA [40,41] which serum levels negatively correlate with cardio-metabolic risk [42]. Therefore, it is likely that most of the beneficial effects could be due, at least in part, to both the decrease in the % of SFA and the increase in the % of both MUFA and PUFA.

The oil mixture treatment for three weeks prevented aging-induced body weight loss. This effect was not mediated by changes in food intake, since the accumulated food intake was significantly lower in old treated rats compared to old rats administered vehicle. The decrease in body weight over the three-week treatment in aged rats treated with vehicle may arise from the stress induced by the gavage administration, since it is reported that aging is associated with an impaired ability to recover from stressful stimuli. Therefore, it is possible that the protective effect of the treatment with the oil mixture preventing body weight loss may be the result of a higher tolerance to stress, as it has been reported after omega-3 PUFA supplementation in men subjected to chronic stress [43]. This is in agreement with the aging-induced increase in adrenal weight, as previously described [44], and the significant reduction after the treatment with the oil mixture.

There is extensive evidence that aging is associated with impaired metabolic function due to increased inflammation and oxidative stress [45,46]. In agreement with this, our results show increased serum levels of total and LDL-cholesterol and triglycerides in old rats, which are the main risk factors for the development of atherosclerosis in aged patients. In addition, old rats also showed increased serum levels of insulin and HOMA-index, and reduced expression of GCK and GSK3β in the liver, which denotes a state of insulin resistance, another important feature of metabolic aging [10]. The aging-induced alterations in both insulin sensitivity and lipid profile were improved by the treatment with the oil mixture, as well as the serum concentrations of IL-6 and the spleen weight that were elevated in old rats. Likewise, other authors have reported that oleic acid has an anti-inflammatory effect [47] and that supplementation with omega-3 fatty acids reduces inflammation both, in the serum [48,49] and in adipose tissue [50], as well as improves insulin sensitivity in humans and in experimental animals [51]. Moreover, the reduced % of SFA and increased % of MUFA may also contribute to improve insulin resistance in old treated rats. In this sense, it is reported that substituting dietary saturated for monounsaturated fat decreases inflammation and fat storage and improves insulin sensitivity [52]. This effect may be explained by the higher oxidation rate of MUFAs compared to SFAs, which results in increased energy expenditure and thus in a lower fat storage [53].

Our results also show that treatment with the oil mixture reduces the expression of pro-oxidant and inflammatory markers in the liver and increases the mRNA levels of antioxidant enzymes, such as GSR and SOD-1. These data are in accordance with a meta-analysis of clinical trials, which shows that omega-3 fatty acid supplementation improves the serum levels of the *antioxidant* enzyme GPx, and reduces the concentrations of the oxidative marker malondialdehyde [54]. It has been proposed that these antioxidant effects are mediated by basal release of nitric oxide and inhibition of NF-κB signaling [55]. The reduced oxidative status in livers from old rats treated with the oil mixture may also be related to the decrease in the % of SFA and increase of MUFA % in serum, since it is reported that an excess of saturated fatty acids leads to severe endoplasmic reticulum and oxidative stress in hepatocytes [56], whereas MUFA exert hepatoprotective effects [27,57].

Our results show that the oil mixture treatment reduces the gene expression of pro-inflammatory markers and attenuates the aging-induced expression of prooxidant enzymes. MUFAs may be responsible, at least in part, for these beneficial myocardial effects, since different studies suggest a cardio-protective role of oleic acid both, in vitro, mitigating TNF-α-induced oxidative stress in rat cardiomyocytes [58], in vivo preventing myocardial injury through reduction of cardiac oxidative stress [59] as well as preventing angiotensin II-induced cardiac remodeling [60]. Likewise, omega-3 fatty acids are also reported to have a clear cardio-protective effect by reducing development of coronary disease [61] and atherosclerosis [62], whereas SFA are reported to exert myocardial deleterious effects [63]. Our results fundamentally align with those of Lennon-Edwards et al. who reported that the antioxidant defense is increased in aged hearts following omega-3 supplementation [64]. However, these authors did not find changes in the expression of inflammatory markers. The discrepancy between our results and those reported by Lennon-Edwards et al. may be explained by differences in rat strains (Wistar vs. Fisher), age of old animals (15 months vs. 24 months) and source of omega-3 fatty acids (fish oil versus algae oil).

In the vascular studies, we found that treatment with the oil mixture did not modify the response to norepinephrine or endothelin-1, but it reduced the aortic contraction to angiotensin II. This agrees with the study of Kenny et al. [65], which reported that supplementation with fish oil reduced the vasoconstriction to angiotensin II (Ang II) but not to norepinephrine or phenylephrine. This inhibitory angiotensin II response may be a link in the protective mechanism of omega-3 fatty acids on the cardiovascular system, since alterations in the renin-angiotensin-aldosterone system (RAAS) are involved in the development of arterial hypertension [66] and atherosclerosis [67]. Paradoxically, our results show that the contraction to direct stimulation of smooth muscle with high potassium concentration was reduced in aged rats and increased by the oil mix treatment. This suggests that aging induces impairment of smooth muscle function, and that this alteration can be attenuated by a supplementation with a mixture of EVOO and AO.

Supplementation with the oil mixture also attenuates aging-induced endothelial dysfunction. These beneficial vascular effects are most likely not mediated by MUFA since oleic acid is reported to inhibit the endothelium-dependent vasodilator response to acetylcholine in rabbit femoral artery [68] and to decrease endothelial nitric oxide synthase (eNOS) activity in cultured endothelial cells [69]. On the contrary, supplementation with omega-3 fatty acids is reported not only to reduce arterial contraction [70,71,72,73], but also to improve endothelium-dependent relaxation [74,75,76,77,78]. Our results agree with those of Farooq et al., which show that omega-3 supplementation reverts the impairment of endothelium-dependent relaxation, the increase in angiotensin AT-1 receptor expression and the oxidative stress induced by aging in rat mesenteric arteries [79]. Thus, the beneficial effects an vascular function are most likely mediated by omega 3 PUFA, although the decrease in SFA may also be involved, since increased levels of SFA are reported to impair endothelium-dependent vasorelaxation in aging [80].

In addition to decreased insulin sensitivity in metabolic tissues, aging is also associated with vascular insulin resistance. In a previous study, we reported that aging is associated with impaired relaxation of rat aorta segments in response to insulin and that this alteration is reversed by a moderate protocol of caloric restriction through activation of the PI3K/Akt pathway and the release of nitric oxide [12]. The present study shows that treatment with the oil mixture has similar effects than caloric restriction, as it also increases the vasodilatation in response in aortic segments, the release of nitric oxide and the expression of p-Akt in arterial tissue in response to insulin. Other studies have also found that omega-3 fatty acids supplementation increases nitric oxide release by activation of the PI3K/Akt pathway in other vascular beds [81]. In addition, the beneficial effects of the oil mixture treatment on aging-induced endothelial dysfunction could also be the result of decrease % of SFA, since SFA are reported to mediate vascular endothelial inflammation and insulin resistance through TLR4-mediated NF-κB and MAPK pathways [82]. Indeed, decreased vascular inflammation and oxidative stress may be the mechanism by which the oil mixture treatment attenuates endothelial dysfunction in aged rats, as it significantly reduced the arterial expression of pro-inflammatory enzymes like COX-2 and prooxidant enzymes like NOX-4. Similarly, other authors have reported that both omega-3 and MUFA fatty acids supplementation exerts antioxidant and anti-inflammatory effects in arterial cells or tissue [79,83].

Overall, these results suggest that supplementation with an oil mixture, composed by EVOO and AO could be a viable procedure to reproduce the beneficial effects of caloric restriction on aging of the cardiovascular system.

It is important to point out that, as the treatment was based on a mixture of AO and EVOO, it is not possible to know exactly which of the specific components are responsible for the beneficial effects. This may be a limitation of the study. However, since the treatment with the oil mixture significantly increased the % of ALA, EPA and DHA in the serum, the beneficial effects on metabolism and vascular function could be due, at least in part, to omega 3 PUFA. However, the possible beneficial effects of EVOO should not be ruled out, since EVOO is also reported to reduce the cardiovascular risk [84] and be useful for the treatment/prevention of metabolic syndrome [85] and endothelial dysfunction [86], due to its anti-inflammatory [21] and antioxidant effects [87,88].

In conclusion, supplementation with a mixture between EVOO and marine AO attenuates aging-induced cardiometabolic alterations, improving vascular function and insulin sensitivity through a reduction of inflammation and oxidative stress. Oil supplements may, therefore, be useful to improve metabolic and cardiovascular health in aging patients.

## Figures and Tables

**Figure 1 antioxidants-09-00483-f001:**
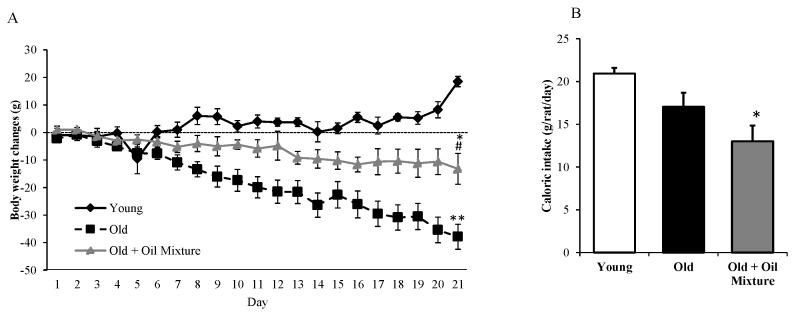
Body weight gain (**A**) and accumulated food intake (**B**) of young rats, old rats and old rats treated 21 days with the oil mixture. Values are represented as mean ± SEM.* *p* < 0.05 vs. Young; ** *p* < 0.01 vs. Young; # *p* < 0.05 vs. Old.

**Figure 2 antioxidants-09-00483-f002:**
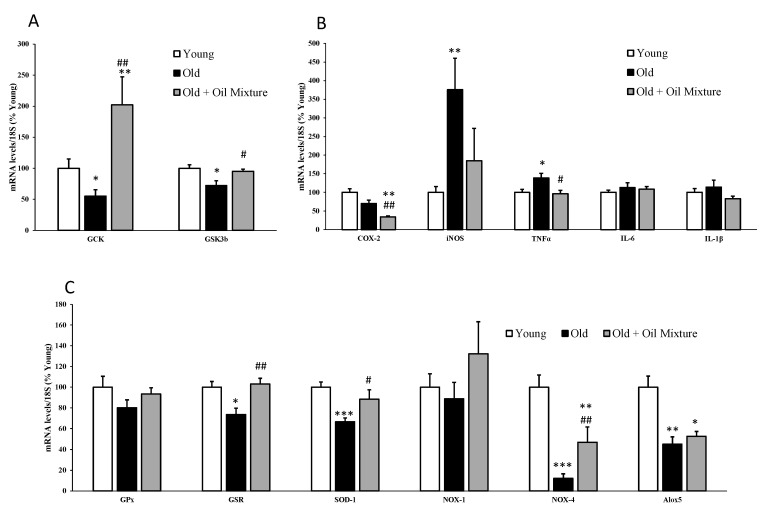
mRNA concentrations of Glucokinase and Glycogen synthase kinase 3β (**A**), ciclooxigenase-2, inducible Nitric Oxide Synthase, Tumor Necrosis Factor α, Interleukin 6 and Interleukin 1β (**B**) and Glutathione Peroxidase, Glutathione Reductase, Super Oxide Dismutase 1, NADPH oxidase 1 and 4, and Lipoxygenase (**C**) in the liver of young rats, old rats and old rats treated 21 days with the oil mixture. Values are represented as mean ± SEM. * *p* < 0.05 vs. Young; ** *p* < 0.01 vs. Young; *** *p* < 0.001 vs. Young; # *p* < 0.05 vs. Old; ## *p* < 0.01 vs. Old.

**Figure 3 antioxidants-09-00483-f003:**
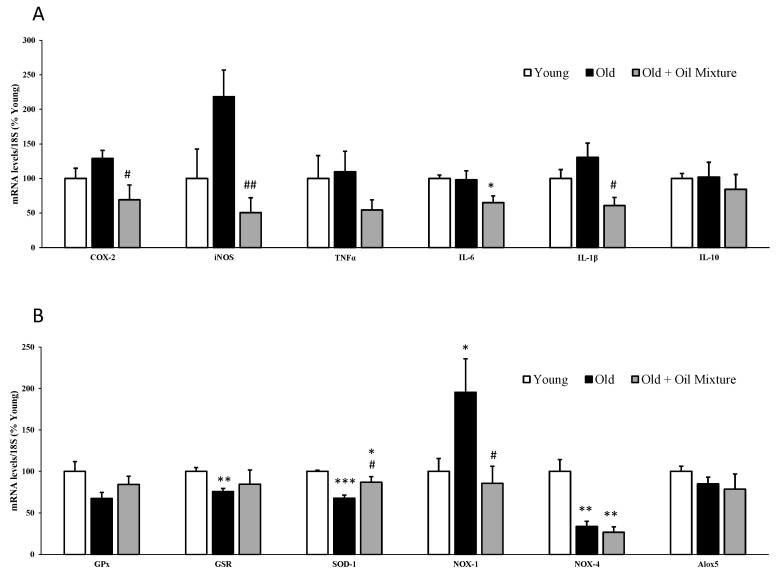
mRNA concentrations of ciclooxigenase-2, inducible Nitric Oxide Synthase, Tumor Necrosis Factor α, Interleukin 6, Interleukin 1β and Interleukin 10 (**A**) and Glutathione Peroxidase, Glutathione Reductase, Super Oxide Dismutase 1, NADPH oxidase 1 and 4, and Lipoxygenase (**B**) in the heart of young rats, old rats and old rats treated 21 days with the oil mixture. Values are represented as mean ± SEM. * *p* < 0.05 vs. Young; ** *p* < 0.01 vs. Young; *** *p* < 0.001 vs. Young; # *p* < 0.05 vs. Old; ## *p* < 0.01 vs. Old.

**Figure 4 antioxidants-09-00483-f004:**
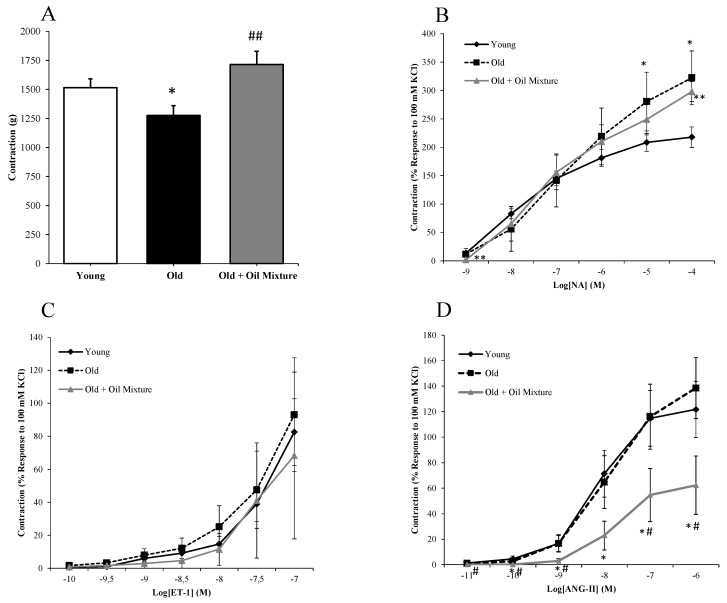
Contraction of abdominal aortic segments to potassium chloride 100 mM (**A**), norepinephrine (**B**), endothelin-1 (**C**) and to angiotensin-II (**D**) of young rats, old rats and old rats treated 21 days with the oil mixture. Values are represented as mean ± SEM. * *p* < 0.05 vs. Young; ** *p* < 0.01 vs. Young; # *p* < 0.05 vs. Old; ## *p* < 0.01 vs. Old.

**Figure 5 antioxidants-09-00483-f005:**
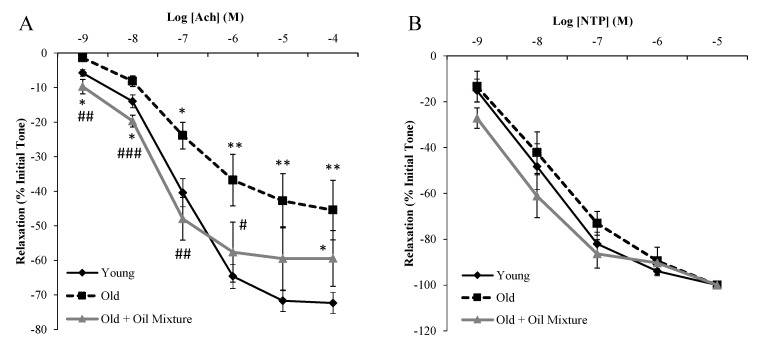
Relaxation of thoracic aortic segments to acetylcholine (**A**) and to sodium nitroprusside (**B**) of young rats, old rats and old rats treated 21 days with the oil mixture. Values are represented as mean ± SEM. * *p* < 0.05 vs. Young; ** *p* < 0.01 vs. Young; # *p* < 0.05 vs. Old; ## *p* < 0.01 vs. Old; ### *p* < 0.001 vs. Old.

**Figure 6 antioxidants-09-00483-f006:**
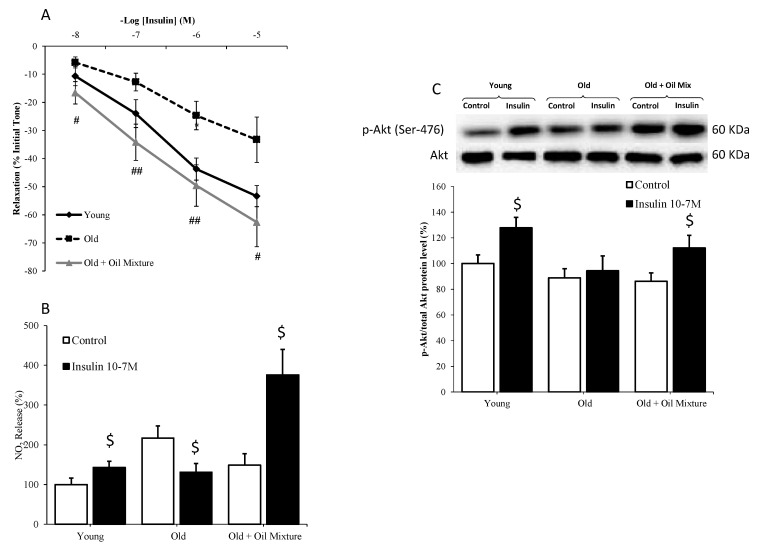
Relaxation of thoracic aortic segments to insulin (**A**), nitrites release from aorta segments to culture medium in presence/absence of insulin (10^−7^ M) (**B**) and ratio between protein levels of p-Akt and Akt in aorta (**C**) of young rats, old rats and old rats treated 21 days with the oil mixture. Values are represented as mean ± SEM. # *p* < 0.05 vs. Old; ## *p* < 0.01 vs. Old; $ *p* < 0.05 vs. Control.

**Figure 7 antioxidants-09-00483-f007:**
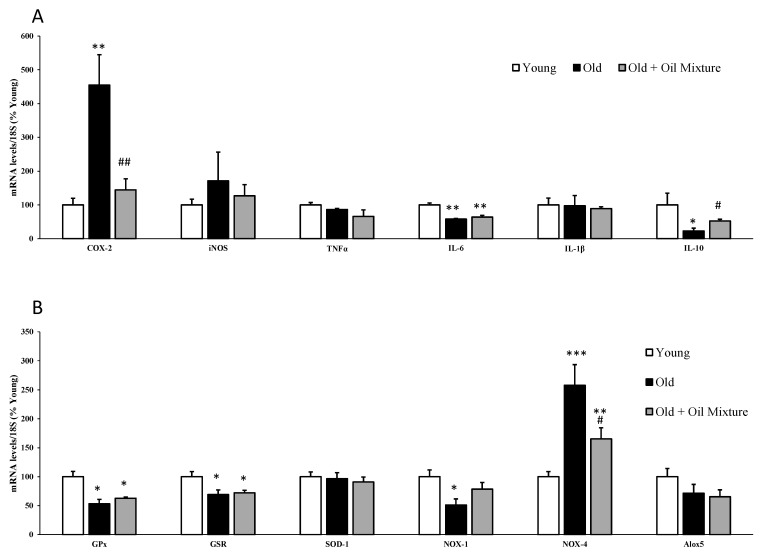
mRNA concentrations of ciclooxigenase-2, inducible Nitric Oxide Synthase, Tumor Necrosis Factor α, Interleukin 6, Interleukin 1β and Interleukin 10 (**A**) and Glutathione Peroxidase, Glutathione Reductase, Super Oxide Dismutase 1, NADPH oxidase 1 and 4, and Lipoxygenase (**B**) in the aorta of young rats, old rats and old rats treated 21 days with the oil mixture. Values are represented as mean ± SEM. * *p* < 0.05 vs. Young; ** *p* < 0.01 vs. Young; *** *p* < 0.001 vs. Young; # *p* < 0.05 vs. Old; ## *p* < 0.05 vs. Old.

**Table 1 antioxidants-09-00483-t001:** Organ weights of young rats, old rats and old rats treated with the oil mixture.

	Young	Old	Old + Oil Mixture
Heart (mg)	1503.8 ± 78.8	1891.3 ± 55.6 **	1530.0 ± 235.0 ^#^
Epidydimal visceral adipose tissue (mg)	9843.1 ± 1075.2	20,475.4 ± 1451.2 ***	20,377.2 ± 1231.4 ***
Lumbar subcutaneous adipose tissue (mg)	4992.4 ± 787.5	27,905.7 ± 6193 **	22,955.9 ± 4816.7 ***
Subscapular brown adipose tissue (mg)	496.3 ± 50.0	810.8 ± 119.4 *	723.2 ± 107.8 *
Periaortic adipose tissue (mg)	180.0 ± 26.0	307.9 ± 47.7 *	303.3 ± 56.4 *
Kidneys (mg)	2530.2 ± 50.2	3298.7 ± 194.2 **	2648.7 ± 204.2 ^#^
Suprarenal glands (mg)	61.9 ± 2.4	86.0 ± 12.6 *	54.8 ± 2.3 ^#^
Liver (mg)	13,465.0 ± 425.6	14,886.9 ± 566.9 *	13,481.8 ± 442.9 ^#^
Spleen (mg)	695.7 ± 20.8	1132.2 ± 61.5 ***	923.2 ± 69.0 **^#^

Data are represented as mean value ± SEM; *n* = 5–11 samples/group. * *p* < 0.05 vs. Young; ** *p* < 0.01 vs. Young; *** *p* < 0.001 vs. Young; # *p* < 0.05 vs. Old.

**Table 2 antioxidants-09-00483-t002:** Lipid profile, hormone concentrations and inflammatory parameters in the serum of young rats, old rats and old rats treated with the oil mixture.

	Young	Old	Old + Oil Mixture
Glycemia (mg/dL)	90.7 ± 3.8	72.4 ± 11.1	81.4 ± 4.4 *
Total Lipids (mg/dL)	853 ± 63	1051 ± 37 *	812 ± 97.7 ^#^
Triglycerides (mg/dL)	97.6 ± 13.5	158.5 ± 36.4 *	73.2 ± 11.8 ^#^
Total Cholesterol (mg/dL)	135.1 ± 15.3	199.3 ± 13.9 *	113.8 ± 9.8 ^###^
LDL-cholesterol (mg/dL)	28.8 ± 2.7	47.8 ± 2.6 **	21.3 ± 1.8 *^###^
HDL-cholesterol (mg/dL)	15.7 ± 0.6	13.4 ± 2.2	10.3 ± 1.2 ***
Insulin (ng/mL)	17.7 ± 5.2	81.3 ± 32.2 **	20.6 ± 2.8
HOMA-Index	1.75 ± 0.3	13.1 ± 5.4 *	3.9 ± 0.5
Leptin (ng/mL)	11.82 ± 1.4	30.2 ± 5.9 **	37.2 ± 1.7 ***
Adiponectin (mg/dL)	67.1 ± 6.7	108.9 ± 4.3 ***	101.7 ± 11.6 **
Interleukin-6 (pg/mL)	135.6 ± 7.1	188.7 ± 24.1 *	127.0 ± 19.0 ^#^
TNFα (pg/mL)	0.1 ± 0.1	1.8 ± 0.9 *	0.1 ± 0.1

Data are represented as mean value ± SEM; *n* = 5–11 samples/group. * *p* < 0.05 vs. Young; ** *p* < 0.01 vs. Young; *** *p* < 0.001 vs. Young; # *p* < 0.05 vs. Old; ### *p* < 0.001 vs. Old.

**Table 3 antioxidants-09-00483-t003:** Fatty acid proportion (%) of young rats, old rats and old rats treated with oil mixture.

	Young	Old	Old + Oil Mixture
SFA	25.0 ± 0.8	28.4 ± 0.4 **	26.9 ± 0.5 ^#^
MUFA	49.6 ± 0.6	42.0 ± 2.2 **	46.3 ± 0.5 **^#^
PUFA	15.2 ± 2.5	16.1 ± 1.2	20.9 ± 1.4 *^#^
Palmitic Acid (C16:0)	8.28 ± 0.3	9.92 ± 0.7 *	8.84 ± 0.7
Palmitoleic Acid (C16:1)	3.52 ± 0.5	5.47 ± 0.5 *	5.74 ± 0.5 **
Stearic Acid (C18:0)	13.2 ± 0.6	13.0 ± 0.5	12.3 ± 0.2
Oleic Acid (C18:1)	47.7 ± 0.7	40.8 ± 1.8 **	45.1 ± 0.7 *^#^
Linoleic Acid (C18:2)	1.8 ± 0.2	1.1 ± 0.4	1.5 ± 0.2
ALA (C18:3)	3.42 ± 0.2	2.7 ± 0.1 **	3.3 ± 0.4
DHA (C22:6n-3)	7.1 ± 1.5	6.9 ± 0.7	9.5 ± 0.7 ^#^
EPA (C20:5n-3)	4.7 ± 1.2	6.5 ± 0.5	8.2 ± 0.7 *^#^

Data are represented as mean value ± SEM; *n* = 5–11 samples/group. * *p* < 0.05 vs. Young; ** *p* < 0.01 vs. Young; # *p* < 0.05 vs. Old. SFA = saturated fatty acids; MUFA = monounsaturated fatty acids; PUFA = polyunsaturated fatty acids; ALA = alpha-linolenic acid; DHA = docosahexaenoic acid; EPA = eicosapentaenoic acid.

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
