# Peer review of "A Mixture of Algae and Extra Virgin Olive Oils Attenuates the Cardiometabolic Alterations Associated with Aging in Male Wistar Rats"

_antioxidants, 2020, doi:10.3390/antiox9060483_

Round 1

Reviewer 1 Report

The manuscript of Gonzalez-Hedström et al analyzes the effect of a mixture of algae oil and extra virgin olive oil on the cardiometabolic alterations associated with aging in rats. Both, olive oil and ω-3 PUFA (present in algae oil) have been associated with beneficial effects in the last years. Nowadays, aging is considered an important cardiovascular risk factor. Thus, the present manuscript analyzes an interesting field. The manuscript shows in an appropriate manner the beneficial metabolic and cardiovascular effects of the mixture of oils used by the authors.

I have two comments:

  1. As the authors say at the end of the manuscript, “it is not possible to know exactly which of specific components are responsible for the beneficial effects, which may constitute a limitation of the study”. It is true. It is an important limitation. Both components have been associated with beneficial effects. It would be nice to know which of both components is the responsible. In this sense, it is necessary to change, in part, the discussion section. Oil mixture is composed by EVOO 75%. The most important component of EVOO is oleic acid (MUFA). However, most of the discussion is centered in the beneficial effects of ω-3 PUFA. It is true that there are also references to MUFA, but taking into account that the authors cannot exclude that the beneficial effects are secondary to the use of EVOO, more aspects considering this fact should be added
  2. Tables and figures are of low quality, at least in my screen. It should be modified.

Author Response

The manuscript of Gonzalez-Hedström et al analyzes the effect of a mixture of algae oil and extra virgin olive oil on the cardiometabolic alterations associated with aging in rats. Both, olive oil and ω-3 PUFA (present in algae oil) have been associated with beneficial effects in the last years. Nowadays, aging is considered an important cardiovascular risk factor. Thus, the present manuscript analyzes an interesting field. The manuscript shows in an appropriate manner the beneficial metabolic and cardiovascular effects of the mixture of oils used by the authors.

We want to thank the reviewer for his/her positive comments about the novelty and interest of the article.

I have two comments:

As the authors say at the end of the manuscript, “it is not possible to know exactly which of specific components are responsible for the beneficial effects, which may constitute a limitation of the study”. It is true. It is an important limitation. Both components have been associated with beneficial effects. It would be nice to know which of both components is the responsible. In this sense, it is necessary to change, in part, the discussion section. Oil mixture is composed by EVOO 75%. The most important component of EVOO is oleic acid (MUFA). However, most of the discussion is centered in the beneficial effects of ω-3 PUFA. It is true that there are also references to MUFA, but taking into account that the authors cannot exclude that the beneficial effects are secondary to the use of EVOO, more aspects considering this fact should be added.

The reviewer raises an important issue regarding the possible role of oleic acid in the beneficial cardiometabolic effects in aged rats. We completely agree with this idea and for this reason we state in the first part of the Discussion section that “it is likely that most of the beneficial effects could be due, at least in part, to both the decrease in the % of SFA and the increase in the % of both MUFA and PUFA.” (Line 404-406).

However, it is true that maybe the Discussion section was more focused in the effects of PUFAs than MUFAs. The reason was that even though there are plenty of studies that report the beneficial cardiometabolic effects of olive oil there a fewer which report the effects of oleic acid alone.

In the revised version of the manuscript, ss suggested by the reviewer, we have now inserted new paragraphs and references within the discussion section to highlight the importance of MUFAs in the effects observed in aged rats treated with the oil mixture.

We have added the following paragraphs:

Lines 427-428: Oleic acid has an anti-inflammatory effect (Medeiros-de-Moraes, Goncalves-de-Albuquerque et al. 2018)

Lines 435-437: This effect may be explained by the higher oxidation rate of MUFAs compared to SFAs, that results in increased energy expenditure and thus in a lower fat storage (DiNicolantonio and O'Keefe 2017).

Line 450-455: MUFAs may be responsible, at least in part, for these beneficial myocardial effects since different studies suggest a cardio-protective role of oleic acid both in vitro, mitigating TNF-α-induced oxidative stress in rat cardiomyocytes (Al-Shudiefat, Sharma et al. 2013), and in vivo preventing myocardial injury through reduction of cardiac oxidative stress (Singh, Gari et al. 2020) and preventing angiotensin II-induced cardiac remodeling (Liu, Wen et al. 2020).

Lines 477-481: Supplementation with the oil mixture also attenuates aging-induced endothelial dysfunction. These beneficial vascular effects are most likely not mediated by MUFA since oleic acid is reported to inhibit the endothelium-dependent vasodilator response to acetylcholine in rabbit femoral artery (Davda, Stepniakowski et al. 1995) and to decrease endothelial nitric oxide synthase (eNOS) activity in cultured endothelial cells (Gremmels, Bevers et al. 2015).

Lines 489-491: Thus, the beneficial effects an vascular function are most likely mediated by omega 3 PUFA, although the decrease in SFA may also be involved since increased levels of SFA are reported to impair endothelium-dependent vasorelaxation in aging (Sloboda, Feve et al. 2012).

 Lines 507-509: Similarly, other authors have reported that both omega-3 and MUFA fatty acids supplementation exerts antioxidant and anti-inflammatory effects in arterial cells or tissue (Hart, Gupta et al. 1997, Farooq, Gaertner et al. 2020).

Tables and figures are of low quality, at least in my screen. It should be modified.

Tables and Figures with higher resolution have been uploaded.

Reviewer 2 Report

The manuscript is very interesting, dealing with the mixture of two oils, extra virgin olive oil (EVOO), and marine algae oil (AO). Both oils exert beneficial health effects, and the proposed combination is particularly interesting since the antioxidative compounds from EVOO protect ω-3 PUFAs from AO, which are highly susceptible to oxidation.  The authors in this paper for the first time analyze the possible beneficial effect of the combined treatment for the treatment of the cardiometabolic alterations associated with aging and conclude that supplementation with the oil mixture attenuates aging-induced cardiometabolic alterations, through a reduction of inflammation and oxidative stress.

The manuscript is well written, the methods and results are properly described and comprehensively discussed.

According to my opinion, the manuscript should be accepted for publication.

Minor comments:

The standard error (SEM) should be denoted in the same way as the mean, with the same number of significant figures. In Table 1, for example, epidydimal visceral adipose tissue; and in Table 3, for example, linoleic acid.

Probability value, p, should be written in italic.

Abbreviation S.E.M. should be explained.

Author Response

The manuscript is very interesting, dealing with the mixture of two oils, extra virgin olive oil (EVOO), and marine algae oil (AO). Both oils exert beneficial health effects, and the proposed combination is particularly interesting since the antioxidative compounds from EVOO protect ω-3 PUFAs from AO, which are highly susceptible to oxidation.  The authors in this paper for the first time analyze the possible beneficial effect of the combined treatment for the treatment of the cardiometabolic alterations associated with aging and conclude that supplementation with the oil mixture attenuates aging-induced cardiometabolic alterations, through a reduction of inflammation and oxidative stress.

The manuscript is well written, the methods and results are properly described and comprehensively discussed.

According to my opinion, the manuscript should be accepted for publication.

We thank the reviewer for the careful revision of our manuscript. We really appreciate the positive comments and his/her willingness to publish the article in this prestigious journal.

Minor comments:

The standard error (SEM) should be denoted in the same way as the mean, with the same number of significant figures. In Table 1, for example, epidydimal visceral adipose tissue; and in Table 3, for example, linoleic acid.

It has been corrected.

Probability value, p, should be written in italic.

It has been corrected.

Abbreviation S.E.M. should be explained.

It has been explained as suggested (Line 246).

Reviewer 3 Report

It was a pleasure to read González-Hedström's manuscript. The authors report the improvement effects from a metabolic and cardiovascular point of view of a treatment with a mixture of algal oil and  extra virgin olive oil in elderly mouse models. The data collected by the authors, inflammatory profile, expression of inflammatory genes, metabolic hormones, body weight, accumulation of body fat, indicate that the proposed treatment improves from systemic degeneration generated by aging. The manuscript is interesting, clear, linear but some critical issues must be clarified for a better exploitation of the work.

  1. The authors should specify the sex of the rats used. It is essential to know the effect of female sex hormones if they had used female rats.
  2. The authors should specify the "young" age of the rats. It is essential to understand the effect of age on the animals used.
  3. The authors should better specify the route of administration of the treatment.
  4. The authors use a generic algae oil extract. They should better specify the species or at least the algal family used for this study. Furthermore, what vegetable part was used for the whole treatment?
  5. More details should be included regarding the varieties of Olea europea
  6. Line 51-53 of the Introduction, the authors should add a phrase indicating the use of molecules / drugs for the reduction of body weight.

Author Response

It was a pleasure to read González-Hedström's manuscript. The authors report the improvement effects from a metabolic and cardiovascular point of view of a treatment with a mixture of algal oil and  extra virgin olive oil in elderly mouse models. The data collected by the authors, inflammatory profile, expression of inflammatory genes, metabolic hormones, body weight, accumulation of body fat, indicate that the proposed treatment improves from systemic degeneration generated by aging. The manuscript is interesting, clear, linear but some critical issues must be clarified for a better exploitation of the work.

We want to thank the reviewer for his/her positive comments about the manuscript.

The authors should specify the sex of the rats used. It is essential to know the effect of female sex hormones if they had used female rats.

As suggested by the reviewer, the sex of the animals (male) is now specified in the Material and Methods section (Line 110)

The authors should specify the "young" age of the rats. It is essential to understand the effect of age on the animals used.

Young rats were 3 months old. It is now stated in the  material and methods section (Line 110) as suggested by the reviewer.

The authors should better specify the route of administration of the treatment.

The treatment was administrated by gavage (intragastric tube). It is now specified in the material and methods section (Line 116).

The authors use a generic algae oil extract. They should better specify the species or at least the algal family used for this study. Furthermore, what vegetable part was used for the whole treatment?

The algae oil was extracted from the spp of microalgae Schizochytrium. This important information is now stated in the Material and Methods section (Line 118)

More details should be included regarding the varieties of Olea europea

As stated in the material and methods section, the variety of EVOO used in this study was Cornicabra variety wich contains 80% oleic acid and 63.49 mg/g of secoiridoids (Aceites Toledo S.A., Los Yébenes, Spain)

Line 51-53 of the Introduction, the authors should add a phrase indicating the use of molecules / drugs for the reduction of body weight.

As suggested by the reviewer the following sentences has been inserted in the Introduction (Lines 51-52)

“and the lack of effective pharmacological treatments to decrease body weight”

Round 2

Reviewer 3 Report

The authors perfected their manuscript. Now, in my opinion, the manuscript could be published.